# A Novel Approach to Learning Models on EEG Data Using Graph Theory Features—A Comparative Study

Bhargav Prakash [ID] , Gautam Kumar Baboo [ID] and Veeky Baths *

Cognitive Neuroscience Lab, Department of Biological Sciences, BITS, Pilani-K.K. Birla Goa Campus, Sancoale 403726, Goa, India; f20170157@goa.bits-pilani.ac.in (B.P.); p20130404@goa.bits-pilani.ac.in (G.K.B.)
* Correspondence: veeky@goa.bits-pilani.ac.in; Tel.: +91-832-258-0436

**Abstract:** Brain connectivity is studied as a functionally connected network using statistical methods such as measuring correlation or covariance. The non-invasive neuroimaging techniques such as Electroencephalography (EEG) signals are converted to networks by transforming the signals into a Correlation Matrix and analyzing the resulting networks. Here, four learning models, namely, Logistic Regression, Random Forest, Support Vector Machine, and Recurrent Neural Networks (RNN), are implemented on two different types of correlation matrices: Correlation Matrix (static connectivity) and Time-resolved Correlation Matrix (dynamic connectivity), to classify them either on their psychometric assessment or the effect of therapy. These correlation matrices are different from traditional learning techniques in the sense that they incorporate theory-based graph features into the learning models, thus providing novelty to this study. The EEG data used in this study is trail-based/event-related from five different experimental paradigms, of which can be broadly classified as working memory tasks and assessment of emotional states (depression, anxiety, and stress). The classifications based on RNN provided higher accuracy (74–88%) than the other three models (50–78%). Instead of using individual graph features, a Correlation Matrix provides an initial test of the data. When compared with the Time-resolved Correlation Matrix, it offered a 4–5% higher accuracy. The Time-resolved Correlation Matrix is better suited for dynamic studies here; it provides lower accuracy when compared to the Correlation Matrix, a static feature.

**Keywords:** EEG; emotional states; working memory; depression; anxiety; graph theory; classification; machine learning; neural networks





## 1. Introduction

Electroencephalography (EEG) is a commonly used neuroimaging tool. Its application ranges from clinical capacity such as sleep disorder studies, to seizure detection, to commercial circumstances such as EEG-controlled games [1]. The EEG data is represented as a two-dimensional matrix, which consists of electric potentials on one axis and the electrode number on the other axis. This form of EEG data makes it easy to use in machine learning models [2]. With its high temporal resolution, EEG data can provide information regarding the functional connectivity within the brain, thereby providing a topological understanding of the functioning of the human brain [3]. This is usually carried out by transforming the electrical potentials into a Correlation Matrix [4].

Functional connectivity is time dependent and to understand the functional aspects of the brain under conditions of executive functions and emotional states viz. depressive or anxious, it is vital to study them in terms of networks and the best way to do it, with the help of EEG signals, which have the highest temporal resolution in the field of neuroimaging techniques. At present learning, models use either the properties of the EEG signal such as amplitude, frequency, and event-related potentials as features or graph properties such as centrality measures which are nodal metrics or edge metrics such as shortest path length.

Network analysis and learning models on neuroimaging data have enabled researchers to study the human brain's functional and structural connectivity [5]. Here, graph metrics

are used as features for a deep learning model, apart from the standard spectral and temporal characteristics that are traditionally used [6]. Different static and dynamic features are studied to understand which features are best suited for visual working memory tasks [7]. Both Convolutional Neural Network (CNN) and Recurrent Neural Network (RNN) are tested and validated for their performance on the datasets.

Previous work on emotional states such as depression and anxiety in the space of EEG and machine learning was carried out using signal features such as power or frequency bands [8]. Learning models such as probabilistic, nearest neighbor, neural network, and tree-based have been implemented on DASS scores, here the Random Forest model provided accuracy in classification of three states, i.e., depressive anxious or stressed at 84%, 85%, and 84% [9,10].

A study on clinically depressed patients and normal controls with the implementation of learning models on EEG signals using features such as frequency bands and non-linear features such as detrended fluctuation analysis (DFA), Higuchi fractal, correlation dimension, and Lyapunov exponent provided an 83.3% accuracy while using Logistic Regression [11]. Similarly, visual and verbal working memory studies using EEG have been carried out using event-related potentials (ERPs) and the subsequent construction of functional connectivity of these ERPs [5]. Studies using EEG and deep learning models involve EEG signals broken into smaller windows for training and testing [12]. The high temporal resolution being the nature of EEG signals adds an additional step in curating these smaller datasets for analysis. This step can induce a bias based on cognitive noise between participants. An SVM implementation to classify Schizophrenic patients and healthy controls based on a working memory task yields an accuracy of >74% [13].

Learning models on EEG data recorded during visual short-term memory task included SVM and Random Forest, which used raw EEG signals and the psychometric assessment scores and reaction times which provided an accuracy of approximately 90% [14]. Other implementations of SVM using frequency bands as features on similar psychological tests yield a 98% accuracy [15]. While using ERPs in the time domain, power spectra and eye-tracking as features provided accuracy in the range of 40% to 60% [7].

The intermediate step between EEG signal analysis and functional connectivity analysis is the use of a Correlation Matrix. This has been used for understanding the brain connectivity in the narrow band signals [16]. The drawback to using the matrix is that it does not address the volume conduction problem or explain the association in different frequency bands. Variations of this method have proved to be helpful to understanding the brain connectivity previously [17,18]. In this study, we explore the utility of the same along with a Time-resolved Correlation Matrix for a comparative learning model study. We compare the two correlation matrices and above all the four learning models: Logistic Regression, Random Forest, Support Vector Machine, and Recurrent Neural Networks, which can shed some light on the nature of EEG activity in these emotional and cognitive states. Using the Correlation Matrix provides a non-directed graph. These kinds of graphs aim to understand the interaction between signals. This enables one to understand the dominant influence at a specific time in the brain signals [19]. Here the EEG data for working memory and emotional states are used from a total of 359 (25(DASS21), 122 (Selection Task), 29 (WM-Lab), 27 (Visual-WM+drug), and 156 (Verbal-WM)) participants. Both EEG data and associated psychometric assessment scores are used for the learning model study. Two high-accuracy models, i.e., recurrent neural network and Random Forest, belong to the neural networks method and ensemble methods. Furthermore, two high interpretable models, a kernel-based method- Support Vector Machine and the Logistic Regression model, are examined and compared.

## 2. Materials and Methods

### 2.1. Datasets

In this study, five EEG datasets are used, of which two were recorded in house, and three are from a public database. Among the two recorded in-house, 25 partici-

pants are from Sternberg Visual Working Memory Task, and 29 participants are from the DASS 21 questionnaire (Figure 1) (approved by the Institute Research Ethics Committee (IHEC-40/16-1)) using a 32 Channel(bipolar montage) EGI geodesic system (Appendix A Figure A1). From the OpenNeuro dataset, 122 participants from Probabilistic Selection Task (OpenNeuro Dataset Accession Number: ds003474) is recorded using a 64 channel Synamps system, 156 participants from verbal working memory Task (OpenNeuro Dataset Accession Number: ds003565) is recorded using a 19 channel 10-20 system Mitsar-EEG-202 amplifier, and 27 participants from visual working memory task (OpenNeuro Dataset Accession Number: ds003519) are used. A total of 359 participants' EEG data is used here (Table 1).

| No. | Question | Never | Sometimes | Often | Almost Always | D | A | S |
|---|---|---|---|---|---|---|---|---|
| 1 | I found it hard to wind down | 0 | 1 | 2 | 3 | | | |
| 2 | I was aware of dryness of my mouth | 0 | 1 | 2 | 3 | | | |
| 3 | I couldn't seem to experience any positive feeling at all | 0 | 1 | 2 | 3 | | | |
| 4 | I experienced breathing difficulty (eg. Excessively rapid breathing, breathlessness in the absence of physical exertion) | 0 | 1 | 2 | 3 | | | |
| 5 | I found it difficult to work up the initiative to do things | 0 | 1 | 2 | 3 | | | |
| 6 | I tended to over-react to situations | 0 | 1 | 2 | 3 | | | |
| 7 | I experienced trembling (e.g. in the hands) | 0 | 1 | 2 | 3 | | | |
| 8 | I felt that I was using a lot of nervous energy | 0 | 1 | 2 | 3 | | | |
| 9 | I was worried about situation in which I might panic and make a fool of myself | 0 | 1 | 2 | 3 | | | |
| 10 | I felt that I had nothing to look forward to | 0 | 1 | 2 | 3 | | | |
| 11 | I found myself getting agitated | 0 | 1 | 2 | 3 | | | |
| 12 | I found it difficult to relax | 0 | 1 | 2 | 3 | | | |
| 13 | I felt down-hearted and blue | 0 | 1 | 2 | 3 | | | |
| 14 | I was intolerant of anything that kept me from getting on with what I was doing | 0 | 1 | 2 | 3 | | | |
| 15 | I felt I was close to panic | 0 | 1 | 2 | 3 | | | |
| 16 | I was unable to become enthusiastic about anything | 0 | 1 | 2 | 3 | | | |
| 17 | I felt I wasn't worth much as a person | 0 | 1 | 2 | 3 | | | |
| 18 | I felt that I was rather touchy | 0 | 1 | 2 | 3 | | | |
| 19 | I was aware of the action of my heart in the absence of physical exertion (ex. sense of heart rate) | 0 | 1 | 2 | 3 | | | |
| 20 | I felt scared without any good reason | 0 | 1 | 2 | 3 | | | |
| 21 | I felt that life was meaningless | 0 | 1 | 2 | 3 | | | |
| | | | | | Total | | | |

**Figure 1.** DASS 21 questionnaire example.

DASS 21 questionnaire is a 21 item self-administered test; this test contains seven sets of questions to assess the three emotional states; depressive, anxious, and stressed. A participant responds with a score ranging from 0 to 3, with 0 meaning never and three meaning almost always. Scores for each category are cumulative; a rating between normal to severe is provided at the end of the test. These scores are then used in classifying participants for the training dataset (Figure 1). Preprocessing of the EEG files are carried out on EEGLab toolbox [20] on MATLAB. Here the data is filtered using the Basic filter option; this option uses the "pop_eegfiltnew()" function from MATLAB. The function filters the data using Hamming windowed sinc FIR filter. The filter order/transition band width is estimated with the following heuristic in default mode: transition bandwidth is 25% of the lower passband edge, but not lower than 2 Hz, where possible (for bandpass,

highpass, and bandstop) and distance from passband edge to critical frequency (DC, Nyquist) otherwise. Window type is hardcoded to Hamming. Furthermore, decomposition of data using Independent Component Analysis (ICA) [21], the filtering and ICA is carried out on the MARA toolbox [22], where the option of automatic removal of components is selected. Following which the data is exported as .set files.

**Table 1.** Overview of EEG datasets.

| Sl. No. | Name of the Dataset | EEG Recording System | Acquisition Parameters |
|---|---|---|---|
| 1 | Visual Working Memory (n = 25) | 32 Channel EGI geodesic | impedance < 50 kΩ, 1000 Hz sampling rate, band-pass filter 0.1–70 Hz, 50 Hz notch filter |
| 2 | Visual Working Memory (n = 27) | 64-channel Brain Vision system | 500 Hz sampling rate, Band-pass filter 0.1–100 Hz |
| 3 | DASS 21 Questionnaire (n = 29) | 32 Channel EGI geodesic | impedance < 50 kΩ, 250 Hz sampling rate, band-pass filter 0.1–70 Hz, 50 Hz notch filter |
| 4 | Probabilistic Selection and Depression (n = 122) | 64 Ag/AgCl electrodes Synamps2 system | impedance < 10 kΩ, 500 Hz sampling rate, band-pass filter 0.5–100 Hz |
| 5 | Verbal Working Memory (n = 156) | 19 electrodes 10–20 system Mitsar-EEG-202 amplifier | 500 Hz sampling rate, band-pass filter 1–150 Hz 50 Hz notch filter |

The task, Probabilistic Selection and Depression (public database), has two tests, the Becks Depression Inventory and the State-Trait Anxiety Inventory [23]. The scores of these tests again range from normal to severe. For the Probabilistic Selection Task [24] the participants were administered the Beck Depression Inventory (BDI) and State-Trait Anxiety Inventory (STAI). Here, BDI scores that are less than or equal to 19 are considered zero and greater than or equal to 20 as one; likewise, for STAI scores, equal to and lesser than 55 are considered as zero and greater than or equal to 56 as one. Bad channels and bad epochs were identified using a conjunction of the FASTER algorithm [25] and the pop_rejchan from EEGLab [20] toolbox and were subsequently interpolated and rejected, respectively. The FASTER algorithm has epoch sensitivity of 97.54%, removes 3.1% of the epochs, and has eye blinks sensitivity of 99.07%. Eye blinks were removed following ICA. Data were re-referenced to averaged mastoids.

Visual working memory (in-house recording) is a modified Sternberg working memory task (Designs, 2021), which involves a visual chart that needs to be memorized/committed to memory, followed by tasks to complete based on the recollection of the chart from memory. Preprocessing of the EEG files are carried out on EEGLab toolbox [20] on MATLAB. Here the data is filtered using the Basic filter option. Decomposition of data using ICA [21] is performed, following which the data is exported as .set files. The recollection of the participant is tested by presenting seven questions on the basis of the visual chart: a score of 50% or less is considered as zero and above as one.

Visual Working Memory and Cabergoline (1.25 mg) Challenge [26], here a drug that can improve memory functions and placebo, is administered to a small group of participants. The placebo and drug groups are used for classification. For the Visual Working Memory and Cabergoline [27] challenge data [28], two sessions are carried out for each participant, one with a placebo and the other with the drug. Here the placebo is treated as zero and drug administered session as one. Data was visually inspected for bad

channels to be interpolated and bad epochs to be rejected. Time-frequency measures were computed by multiplying fast Fourier transformed (FFT) power spectrum of single trial EEG data with the FFT power spectrum of complex Morlet wavelets. The end result of this process is the same as time-domain signal convolution.

Finally, verbal working memory (public database) [29] consists of the EEG recorded in a modified Sternberg working memory paradigm with two types of tasks, with mental manipulations (alphabetization), simple retention (TASK), and three levels of load and 5, 6, or 7 letters to memorize (LOAD). When the participant is able to answer greater than 50% of the time in the trial it is considered one and below 50% is considered zero. First, ocular activity artifacts were addressed using ICA using AMICA algorithm [30]. Second, epochs containing artifacts were visually identified and discarded [31]. EEGLab [20] was used for data preprocessing.

Apart from exploring the utility of the Correlation Matrix, a comparison between the data recorded in-house and the public database is carried out using the accuracy of the models. EEG artifacts suppression and removal was conducted in the following two steps.

### 2.2. Computation of Correlation Matrix and Time Resolved Correlation Matrix Using Brainstorm Toolbox

The .set files are then imported onto the Brainstorm toolbox [32]. Here, using the Editor pipeline, the connectivity option is used for computing the Correlation Matrix and Time-resolved Correlation Matrix.

In this connectivity analysis, the following points are considered:

- The EEG sensors data is used from each of the datasets.
- Trail based data is drawn on.
- Full networks are calculated.
- In terms of temporal resolution, both static and dynamic are studied.
- The output data has a 4-D structure: Channels X Channels X Frequency Bands X Time.

#### 2.2.1. Correlation Matrix Computation

The editor pipeline for computing the Correlation Matrix, the connectivity option for Correlation Matrix, provides three option windows as follows: Input options, Process options, and finally Output options. These options are presented in a nutshell below:

1.  The Input option has three input fields, namely:

    (a) Time window.
    (b) Sensors types or names.
    (c) Checkbox to include bad channels.

2.  The process option has a checkbox to allow for computing the scalar product instead of correlation.
3.  Finally, output options, which has two checkboxes: (1) for saving individuals' results (one file per input file) and (2) for saving the average connectivity matrix (one file).

#### 2.2.2. Time Resolved Matrix Computation

In case of Time resolved matrix, the editor has two main options: Input options and Process Options which are described briefly below.

1.  Input option has three input fields:

    (a) Time window.
    (b) Sensor types or names and a checkbox to include bad channels.

2.  Process option has:

    - Estimation window length (350 ms).
    - Sliding window overlap (50%).
    - Estimator options: computing the scalar product instead of correlation.
    - Output configuration (enables addition of comment tag).

*2.3. Methods*

2.3.1. Data Processing

Given the sensitivity of the EEG signals, it is imperative to preprocess them before any other analysis of the data is carried out. Therefore, the EEG data is filtered to remove line noise (50 Hz), band-pass filters, removal of bad channels, and artifact removal (please refer to Section 4.1 datasets for details), and this data is converted into a Correlation Matrix (NxN), each matrix corresponds to each EEG session file and time resolved Correlation Matrix (NxN), with a 50% sliding window overlap and 350 ms window length. The matrix is square and symmetrical, where each cell entry is the correlation between any two EEG electrodes; these operations are carried out on the Brainstorm package [33] on MATLAB.

Principal Component Analysis (PCA) is carried out with the help of scikit-learn [34] before using the data as input. This helped in two ways: it reduced the dimension of data while preserving the features and is a standard method for removing multicollinearity.

*2.4. Learning Models*

After preprocessing and feature extraction of the original EEG data, the Correlation Matrix (feature) is used as input to different classifiers, including traditional machine learning algorithms and neural networks tuned in line with our data. The models used are Logistic Regression, Random Forest, Support Vector Machine, and Recurrent Neural Networks (RNN) to classify the EEG data. The performance evaluation of the different classifiers is examined using a confusion matrix, whose components are T.P., TN, F.P., and F.N. Further, the accuracies are calculated using these measures, using the formula:

$$Accuracy = (TP + TN)/(TP + FP + TN + FN) * 100 \qquad (1)$$

T.P.: True Positives T.N.: True Negatives F.P.: False Positives F.N.: False Negatives.

Overfitting/underfitting: In this study the problem of overfitting did not pose an obstacle in these datasets. Hence, the results were not unusually accurate. Regularisation factors are applied to reduce overfitting. Finally for underfitting, varying the hyperparameters over a large range is carried out and the best fitting set of values is appropriated.

2.4.1. The Logistic Regression Model (LR)

A Logistic Regression model with Gaussian Kernel and Laplacian Prior is used for classification. The Gaussian kernel optimizes the separation between data points in the transformed space obtained in preprocessing, while the Laplacian Prior enhances the sparseness of learned L.R. regressors to avoid overfitting [8]. A multinomial L.R. model where the probability that an input feature xi belongs to class k is given by:

$$p(y_i = k | x_i, w) = \frac{exp(w^{(k)}h(x_i))}{\sum_{k=1}^{K} exp(w^{(k)}h(w_i))}, \qquad (2)$$

$x_i$: feature vector
$k$: class
$h(x_i)$: linear transformation function of $x_i$
$w$: logistic regressors.

2.4.2. Support Vector Machine (SVM)

Apart from the application of SVM on EEG data, implementation of SVM on MRI data to classify between major depressive disorder and bipolar disorder provided accuracy up to 45% to 90% [35]. The main reason behind using SVM is to leverage its relatively less computational power to produce a significant accuracy and to reduce possible redundant information (which is very common in EEG datasets) residing in the data. The input data is mapped to a higher dimensional vector space using a linear kernel function to find a hyperplane for classification (Figure 2).

$$w * z - b = 0 \qquad (3)$$

$w$: normal vector

$b$: bias of separation of hyperplane.

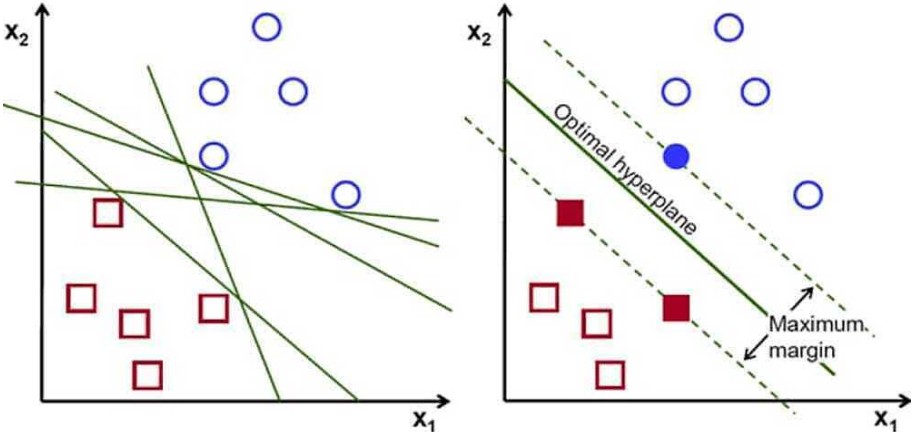

**Figure 2.** Support Vector Machine algorithm with the construction of different hyperplanes that separates the different classes. The most optimal hyperplane is the one that maximizes this separation.

### 2.4.3. Random Forest (RF)

A Random Forest classifier (Figure 3) that uses an ensemble learning approach towards prediction is used. R.F. classifier works in a similar way as the decision tree classifier, only with an ensemble learning approach added to it. The first step is the creation of many random decision trees, each predicting a particular class according to the features given to it. Once each tree predicts a class, voting is carried out to take into consideration the final class according to a majority. The output is then the class that has the majority voting.

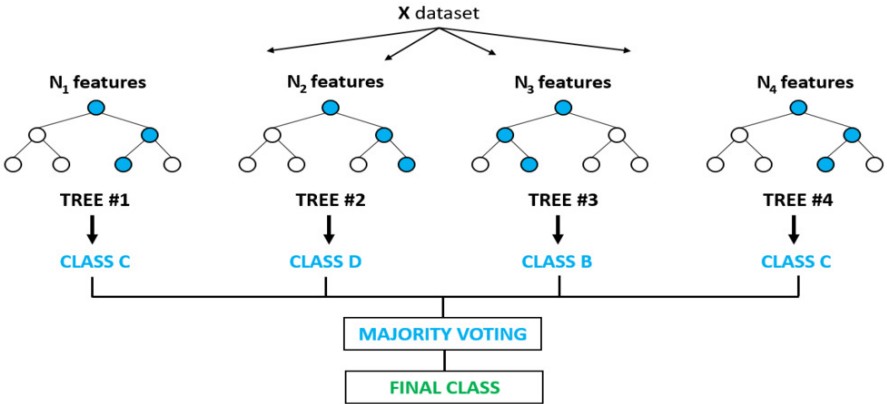

**Figure 3.** Ensemble method implemented by the Random Forest Algorithm. The ensemble consists of different trees fitted on the data with a range of hyperparameters. The tree which fits the data most optimally is then chosen by the algorithm by majority voting method.

### 2.4.4. Recurrent Neural Network (RNN)

Previous work on the implementation of neural networks on EEG signals has been fruitful, which provided accuracy in the range of 81% to 94% [36]. RNN was a good model for studying both working memory [37,38] and emotional state [39] EEG data when compared to other models such as SVM or deep belief networks [40]; on that note the following RNN model is implemented. The RNN is implemented through a Long Short Term Memory (LSTM) model [6,41], producing exemplary results on sequential data, such

as EEG data. A sequential model is used to build the LSTM, which is a linear stack of layers. The first layer is an LSTM layer with 256 memory units, and it defines the input shape. This is done to ensure that the next LSTM layer receives sequences and not just randomly scattered data. The next layer is a Dense layer with a "sigmoid" activation function. A dropout layer is applied after each LSTM layer to avoid overfitting of the model. The model is then trained and monitored for validation accuracy using loss as "binary cross-entropy", optimizer as "adam", and metrics as "accuracy" (Figure 4).

$$H(q) = -1/N \sum y_i * log(p(y_i)) + (1 - y_i) * log(1 - p(y_i)) \tag{4}$$

$H(q)$: binary cross entropy
$p(y_i)$: probability of belonging to class $y_i$.

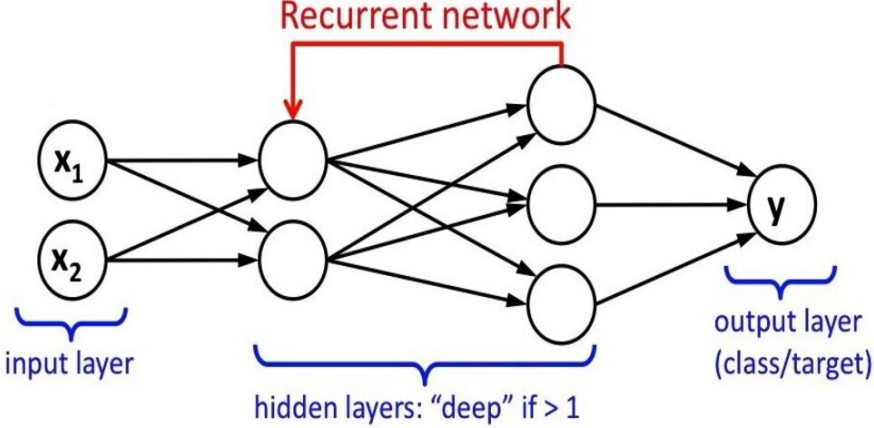

**Figure 4.** Recurrent Neural Network (RNN), representing the skeleton on which every RNN is built. The output of each layer acts as the input to the next and modifies the hyperparameters of the layer in each epoch, thus implementing the learning part of the algorithm.

## 3. Results

The performance of RNN classifiers shows up to 94.50% and 88.64% accuracies for each of the working memory tasks, which outperforms most of the previous works reviewed. The performance of R.F. and L.R. classifiers are relatively sub-par compared to RNNs but still comparable to previously obtained results. The poor performance of SVMs highlights the shortcomings of the method adopted in this study in algorithms that are sensitive to the dimensions of the data. The impressive performance of RNNs can be attributed to their innate ability to extract correlated features, which are not visible in traditional statistical methods, within the data with the help of their stacked networks and activation functions. The standard performance of R.F. and L.R. algorithms highlights the validity of the method adopted in this study and the enormous scope it provides for further improvement.

Further, the data from the public database provides higher accuracy (Tables 2 and A2) in all four models when compared to the in-house data (Tables 3 and A3). On average there seems to be a difference of 40–60% accuracy between the two groups.

**Table 2.** Accuracy of Classifying Emotional States from the Probabilistic Selection Task Data.

| Emotional State/ Learning Model Accuracy | Logistic Regression | Random Forest | SVM | RNN |
|---|---|---|---|---|
| Depression | 71.33% | 73.46% | 61.78% | 88.64% |
| Anxiety | 64.56% | 78.66% | 65.27% | 80.75% |

**Table 3.** Classifying Emotional States from the DASS 21 Data.

| Emotional State/ (% Accuracy of Model) | Logistic Regression | Random Forest | SVM | RNN |
|---|---|---|---|---|
| Depression | 35.06% | 28.60% | 27.60% | 34.75% |
| Anxiety | 28.40% | 34.45% | 30.85% | 38.85% |
| Stress | 31.10% | 33.20% | 31.70% | 36.40% |

## 4. Discussion

The current adaptation of learning models for studying brain connectivity with EEG dataset involves feature extraction from the signal itself, such as the power spectral density or event-related potentials (ERP). Or, when the EEG data is transformed into a graph, graph features such as nodal/edge metrics need to be calculated first before the machine learning process (which is done for some of the datasets that were obtained from OpenNeuro [26,42]). These steps require a dossier containing the experimental paradigms, the brain regions involved in the testing condition, specifics of band frequencies, Transition Frequency, and ERPs. This entails added/newer steps to various stages of the data processing. This translates to increased time and not to mention the myriad of statistical analyses that need to be carried out.

Methods such as phase coherence, phase locking value, or pairwise phase consistency which transform the EEG data to a matrix form for network construction, require adding steps to the analysis which translates to more time spent [19]. These methods and the convenience or inconvenience a method can add to the analysis pipeline takes the analysts on a puzzling path to addressing them with more tools to appreciate or tackle the unexpected observations or results.

Functional Connectivity using EEG data can be done on the basis of frequency, time domain, or phase characteristics of the signal. This can further be categorized as static or dynamic. The various methods under each of these categories have their own advantages and disadvantages, followed by tools/methods that can strengthen or weaken the said methods of functional connectivity analysis. Some of the common challenges with EEG-functional connectivity studies include (1) The common Reference Problem: the use of unipolar reference scheme tends to provide false coherence whereas (2) the bipolar reference scheme or (3) unipolar with separate reference address the problem inherent with unipolar reference schemes [19] (note: The EEG datasets used in this study used bipolar reference schemes).

To address the obstacle of the signal to noise (SNR) ratio, the impedance during the recordings is maintained and monitored as stringently as possible. Although, the best practice to address the SNR problem would be to use stratification methods or a suitable post-hoc method. The sample bias problem is addressed by using each of the trials as an input to the learning models. To circle back, in this study the main aim is to evaluate the performance of the four learning models.

To circle back, in this study the main aim is to evaluate the performance of the four learning models.

Implementation of learning models on imaging data to study emotional states provided reliable results in the past [43]. With the use of both high accuracy (RNN and R.F.) and high interpretability (SVM and L.R. model), we can look for non-linear relationships, non-smooth relationships, and well-defined relationships.

Comparison of learning models on similar paradigm EEG data helps with functional connectivity study. Here, it is demonstrated that a Correlation Matrix can be used in learning models and provides exemplary accuracy. Furthermore, it yielded higher accuracy rates with well-structured data obtained in a controlled environment, as with the working memory tasks, indicating superior discriminatory performances when assessing mental tasks. In addition, the present study is discriminatory towards poorly collected and insufficient data.

From running the classification models on both types of datasets: correlation and Time-resolved Correlation Matrices, we find that the two classification models: Random Forest Classifier and RNN classifier, perform relatively better when the correlation is not time-resolved. The performance dips across both the Verbal Memory and Working Memory datasets for time-resolved correlation. This provides scope for further research as to why dynamic methods may not be a better fit for Neural Networks and Decision Tree based classification models.

This study sheds some light on brain networks when studying emotion or executive functions such as working memory. By using the Correlation Matrix as such, this enables us to study the brain activity as a complete network and not sub-networks or brain regions [44]. Another underlying quality of the participants is their linguistic abilities. The data collected in-house had participants who were at least bilingual, of the 59 participants only 5 were bilinguals and the rest either had adequate knowledge of a third language or fourth. Similarly, the datasets of the probabilistic selection task [45] and the verbal working memory task [42] consist of participants who know English as well as Japanese (selection task) and Russian (working memory). The results from this study and the need to understand the bilingual/multilingual neurocognition [46] of individuals necessitate a deeper study into the role of language on emotional states and working memory. A comprehensive study into the static/dynamic metrics under the three categories of time, frequency, and phase would help in understanding which methods and which parameters can work best for a particular experimental paradigm.

The engagement of participants in the five experimental paradigms is by nature dynamic states of EEG activity. With this in mind, the use of a Time-resolved Correlation Matrix is explored alongside the Correlation Matrix, both features of the time domain of the signal. Since the results indicate the use of the dynamic feature is most suitable for such cognitive states, it sets the stage to explore the other static and dynamic features of trial-related EEG data. At this stage, this investigation provides a step for exploring the possibility of using these features as markers for the cognitive footprints of psychopathologies such as memory and emotional state deficits.

The results indicate that using graph metric for dynamic (Correlation Matrix) features is optimum. Computerized administration of the test rules out pressure to perform or dishonesty.

### 4.1. Limitations

Given that both the positive and negative lag indicates an influence in the network, the bi-directional interactions that could be occurring are beyond the scope of the current study [19]. In this regard, it is to be noted that in EEGLab, filtering for connectivity analysis can be carried out using the "Basic Filter (legacy)". This applies the filter forward and then again backward to ensure that phase delays introduced by the filter are nullified. The "causal filter" (part of the "Basic Filter (legacy)") is only applied forward, so phase delays might be introduced and not compensated for. However, causal relationships are preserved. This introduces the problem of phase distortion. In this study, the common input problem is not dealt with as it would increase the number of steps involved in the preprocessing of the EEG data and also increase the run time of the pipeline. The current study does not have the resolution to examine the salience, executive, and task-related networks or provide a distinction between the three [44].

Although RNN and Random Forest models provide high accuracy, both these methods have longer run times when compared to the other two. In the current study, the lack of defined healthy control groups across the datasets can be addressed, which can help improve the accuracy of the models. This imbalance can be addressed using larger data and a robust learning model [47].

Single trials in the case of the in-house dataset and using DASS 21 for the first time as a computerized test and EEG could explain the lower accuracy across the models associated with this data. This also applies to the visual working memory data recorded in the lab.

The EEG data is collected from four different EEG acquisition systems with five different acquisition parameters. Furthermore, the experimental paradigms are dissimilar along with the distribution of participants among the two main study areas, i.e., emotional states (n = 176) and working memory (n = 183) which is uneven.

Using graph features on the EEG data is time consuming because graph features can range from nodal metrics to local/global network characteristics that need to be considered features. Simultaneously cherry-picking graph metric(s) can introduce a bias that has to be considered in the study and addressed at a later point with defined statistical analysis.

## 5. Conclusions

The time-series nature of the EEG data, which is an effective form of neuroimaging data for studying the functional connectivity of the brain, is studied for its utility in a machine learning environment. Although this is not a first of its kind, the use of the Correlation Matrix/Time-resolved Correlation Matrix makes it one. The previous work on implementing learning models on EEG data consists of using features from the signal processing field. These studies provide insight into the possible electrical activity of each lobe(s) associated with the behavior. However, they fall short while explaining the possible functional connectivity between the regions of the brain or the whole brain. Using such EEG datasets recorded on the working memory and emotional state assessment paradigms, a preliminary comparison of the different EEG acquisition systems and acquisition parameters is attempted.

The application of the Correlation Matrix can be implemented as a first step into choosing the appropriate learning model for studying the emotional or working memory EEG data. This study reveals that using a Correlation Matrix instead of a Time-resolved Correlation Matrix even under trail-based EEG data is a better-suited input for learning models when compared to a dynamic feature such as the Time-resolved Correlation Matrix. This brings us to the experiments themselves.

The memory tasks and psychometric assessment tests—BDI, STAI, and DASS 21— involve different brain regions, given that they have to be functionally connected to respond to the questions in these tests. This study provides a basis for studying the cognitive footprints for memory deficits, depression, anxiety, and stress. Further, it is observed that RNN performs the best compared with the other three models implemented in this study.

**Author Contributions:** Conceptualization, G.K.B. and V.B.; methodology, B.P.; software, B.P.; validation, G.K.B. and B.P.; formal analysis, G.K.B. and B.P.; investigation, G.K.B. and B.P.; resources, V.B.; writing—original draft preparation, G.K.B. and B.P.; writing—review and editing, V.B.; visualization, B.P. and G.K.B.; supervision, V.B.; project administration, V.B.; funding acquisition, Veeky Baths. All authors have read and agreed to the published version of the manuscript.

**Funding:** We thank the Department of Science and Technology, Government of India for the grant (SR/CSRI/50/2014(G)) and Department of Biological Sciences, BITS, Pilani-K.K. Birla Goa Campus for the infrastructure support.

**Institutional Review Board Statement:** The study was conducted according to the guidelines of the Declaration of Helsinki and approved by the Institutional Ethics Committee of Birla Institute of Technology and Science, Pilani (IHEC-40/16-1).

**Informed Consent Statement:** Informed consent was obtained from all participants involved in the study. Written informed consent has been obtained from the participants to publish this paper.

**Data Availability Statement:** The data presented in this study are openly available in OpenNEURO repository-

- EEG: Visual Working Memory + Carbergoline Challenge Dataset
  https://openneuro.org/datasets/ds003519/versions/1.1.0
  DOI:10.18112/openneuro.ds003519.v1.1.0,
- EEG: Probabilistic selection task and Depression Dataset
  https://openneuro.org/datasets/ds003474/versions/1.1.0
  DOI:10.18112/openneuro.ds003474.v1.1.0,
- VerbalWorkingMemory Dataset
  https://openneuro.org/datasets/ds003655/versions/1.0.0
  DOI:10.18112/openneuro.ds003655.v1.0.0.

  And In-house datasets can be accessed here

- DASS 21 Questionnaire EEG recordings-https://tinyurl.com/cvd729p8 and
- Working Memory EEG recordings-https://tinyurl.com/2z6ms7p6

**Conflicts of Interest:** The authors declare no conflict of interest.

## Appendix A

*Reproduction of the Research Shown*

1. in-house EEG datasets please follow the steps provided below

   - import the files to EEGLab on MATLAB
   - filter the files using the MARA Toolbox using band-pass filter 0.1–70 Hz and 50 Hz notch filter
   - please select automatic ICA rejection
   - export the files as .set format
   - import the files (create a study for each dataset) on to brainstorm toolbox on MAT-LAB.
   - use the connectivity editor for computing Correlation Matrix

2. For the OpenNEURO datasets

   - import the files(create a suitable study protocol for each dataset) on to brainstorm
   - use the connectivity editor for computing Correlation Matrix
   - In case the files on OpenNEURO are RAW files, follow the steps provided on the readme file for preprocessing of the EEG recordings.

Please follow the link-https://github.com/bhargavPrak/eeg_classification, accessed on GitHub (accessed on 26 August 2021) for brief description of the skeleton of the machine learning models implemented in this study.

Note: Please refer to the articles for each of the OpenNEURO datasets, since each of them have implemented specific and distinct preprocessing techniques which were best suited for the experimental paradigm. Any deviation from the methods used would impact the overall accuracy obtained from the machine learning models.

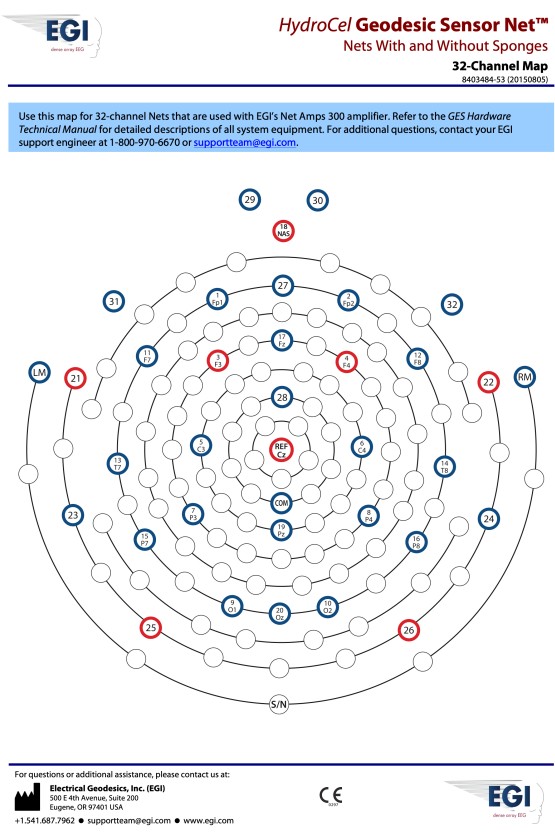

**Figure A1.** EEG Sensor Placement, the signals of each sensor helps in studying the activity of the particular region of the brain. This further helps in functional connectivity studies of the brain.

**Table A1.** Accuracy of Classifying Placebo vs. Drug induced Memory Task conditions.

| Condition | Logistic Regression (% Accuracy) | Random Forest (% Accuracy) | SVM (% Accuracy) | RNN (% Accuracy) |
|---|---|---|---|---|
| Placebo | 73.60 | 80.40 | 73.50 | 90.20 |
| Drug | 71.80 | 81.60 | 76.80 | 92.80 |

**Table A2.** Accuracy of Classifying Verbal Memory Task Conditions 5, 6 or 7 letters.

| | 5 | 6 | 7 |
|---|---|---|---|
| Manipulation | Logistic regression–66.66% Random forest–65.50% SVM–60.15% RNN–75.86% | Logistic regression–59.40% Random forest–69.40% SVM–59.80% RNN–70.40% | Logistic regression–61.10% Random forest–76.70% SVM–54.70.10% RNN–71.50% |
| Retention | Logistic regression–68.70% Random forest–70.60% SVM–55.60% RNN–74.80% | Logistic regression–66.40% Random forest–65.80% SVM–50.20% RNN–70.60% | Logistic regression–63.40% Random forest–68.30% SVM–53.30% RNN–79.60% |

**Table A3.** Participants of Modified Sternberg Working Memory Task.

|  | Logistic Regression (% Accuracy) | Random Forest (% Accuracy) | SVM (% Accuracy) | RNN (% Accuracy) |
|---|---|---|---|---|
| Participant 01 | 12.5 | 37.5 | 28.60 | 12.5 |
| Participant 02 | 25 | 28.30 | 28.60 | 28.60 |
| Participant 03 | 14.30 | 37.5 | 14.30 | 14.30 |
| Participant 04 | 50 | 12.5 | 25 | 25 |
| Participant 05 | 25 | 25 | 25 | 28.60 |
| Participant 06 | 25 | 12.5 | 12.5 | 14.30 |
| Participant 07 | 14.30 | 42.90 | 12.5 | 50 |
| Participant 08 | 12.5 | 25 | 12.5 | 12.5 |
| Participant 09 | 50 | 28.60 | 22.22 | 25 |
| Participant 10 | 75 | 50 | 14.60 | 14.60 |
| Participant 11 | 12.5 | 12.5 | 28.60 | 22.22 |
| Participant 12 | 37.5 | 50 | 11.11 | 12.5 |
| Participant 13 | 28.60 | 14.30 | 25 | 28.60 |
| Participant 14 | 12.5 | 12.5 | 37.5 | 14.30 |
| Participant 15 | 25 | 25 | 37.5 | 25 |
| Participant 16 | 25 | 12.5 | 12.5 | 12.5 |
| Participant 17 | 28.60 | 25 | 50 | 33.33 |
| Participant 18 | 12.5 | 37.5 | 25 | 14.60 |
| Participant 19 | 50 | 12.5 | 37.5 | 25 |
| Participant 20 | 14.30 | 14.30 | 14.30 | 12.5 |
| Participant 21 | 25 | 37.5 | 14.30 | 12.5 |
| Participant 22 | 12.5 | 25 | 22.22 | 14.30 |
| Participant 23 | 14.30 | 25 | 28.60 | 25 |
| Participant 24 | 25 | 12.5 | 12.5 | 28.60 |
| Participant 25 | 50 | 28.60 | 12.5 | 12.5 |

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
