# Peer review of "A Novel Approach to Learning Models on EEG Data Using Graph Theory Features—A Comparative Study"

_2504-2289, doi:10.3390/bdcc5030039_

Round 1

Reviewer 1 Report

To my understanding, your work is mainly a review of the major methods of performing classification of data collected from EEG signals with the purpose      of assessing emotional states of humans. The novelty of  your work is the application of these  methods by forming correlation matrices with the available data, an issue that raises the accuracy of prediction. You have not however  show how these matrices are formed and in the methods section you have to present the mathematical form the methods take  when these matrices are used. Also you have listed a number of figures without making any reference to these figures in  the text. The  reader cannot  easily  understand what each figure is all about.

Author Response

Dear Reviewer,

Big Data Cognitive Computing,

Special Issue- Knowledge Modelling and Learning through Cognitive Networks,

Thank you for giving me the opportunity to submit a revised draft of my manuscript titled “A Novel Approach to Learning Models on EEG Data Using Graph Theory Features-A Comparative Study” to the Special Issue of Big Data and Cognitive Computing titled “Knowledge Modelling and Learning through Cognitive Networks”. We appreciate the time and effort that you have dedicated to providing your valuable feedback on our manuscript. We are grateful for the insightful comments on our paper. We have been able to incorporate changes to reflect most of the suggestions provided by the reviewers. We have highlighted the changes within the manuscript.

Here is a point-by-point response to the reviewers’ comments and concerns. 

To my understanding, your work is mainly a review of the major methods of performing classification of data collected from EEG signals with the purpose of assessing the emotional states of humans.

The novelty of your work is the application of these methods by forming correlation matrices with the available data, an issue that raises the accuracy of prediction.

Comment: You have not however shown how these matrices are formed and in the methods section you have to present the mathematical form the methods take  when these matrices are used.

Response: The description and the form the matrices take have been added “Section 4.2.1 Data processing” see lines 247-252 “Matrics are of the form NxN, where the cell entries are the correlation value between two electrodes”

Comment: Also, you have listed a number of figures without making any reference to these figures in the text. The reader cannot  easily  understand what each figure is all about.

Response: The figures have been referenced in the main text.

Reviewer 2 Report

In this work, the authors applied four different learning models to EEG data from different datasets. 

My major concerns:

1)    Compute a connectivity matrix from EEG data is not trivial and there are in the EEG community a lot of open problems and pitfalls in estimating brain connectivity. No one of these open issues were neither addressed nor discussed in the Introduction and Discussion section.

2)    The EEG pre-processing is qualitatively and not quantitatively described. Since the pre-processed EEG is the input of the authors’ learning models, I am not able to interpret the results. Moreover, it is not possible for the reader to reproduce the authors’ results without having any information about the pre-processing.

Abstract –

Lines 2-3:  Oversimplistic definition of Functional Connectivity. See, for example: http://www.scholarpedia.org/article/Brain_connectivity

Line 5: Which methods were used to estimated static and dynamic connectivity?

Lines 8-9. Please, describe briefly your dataset giving details on acquisition/experimental protocol

Line 9: RNN was not defined before (to define in line 4?)

Lines 13-14: The conclusion statement is not clear to me.

Introduction – 

Line 19 “:” instead “;”
Line 20: EEG data can be seen as a matrix. Please, use more specific language through the manuscript.

Line 21-26: To compute the connectivity matrix from EEG recordings is still an open problem and full of pitfalls. See, for example: doi:

10.3389/fnsys.2015.00175/full

10.1007/s10548-018-0691-2

Line 37. Please, define CNN and RNN

Limitations – 
Among, limitations should be cited that data exploited were recorded with different systems and from different personnel. 

Material and Methods –
Lines 178-179. There are no details on EEG pre-processing, e.g.: kinds of filter, order, if authors performed a zero-phase digital filtering to avoid signal phase distortion. How were artefacts removed? By visual inspection? By ICA? If so, which were the criteria?

Line 192. The EEG pre-processing is not quantitatively described, there are no details. Moreover, it is not described how the correlation matrix was computed and how the volume conduction/field spread problem was addressed 

Author Response

Dear Reviewer,

Big Data Cognitive Computing,

Special Issue- Knowledge Modelling and Learning through Cognitive Networks,

Thank you for giving me the opportunity to submit a revised draft of my manuscript titled “A Novel Approach to Learning Models on EEG Data Using Graph Theory Features-A Comparative Study” to the Special Issue of Big Data and Cognitive Computing titled “Knowledge Modelling and Learning through Cognitive Networks”. We appreciate the time and effort that you have dedicated to providing your valuable feedback on our manuscript. We are grateful for the insightful comments on our paper. We have been able to incorporate changes to reflect most of the suggestions provided by the reviewers. We have highlighted the changes within the manuscript.

Here is a point-by-point response to the reviewers’ comments and concerns.

Reviewer 2.

My major concerns:

Comment: Computing a connectivity matrix from EEG data is not trivial and there are in the EEG community a lot of open problems and pitfalls in estimating brain connectivity. No one of these open issues were neither addressed nor discussed in the Introduction and Discussion section.

Response: Challenges such as signal to noise is addressed in the preprocessing steps of the data; band pass filtering, notch filter rejection of bad epocs and bad channel replacement methods. The problem of volume conduction or the association of different frequencies is not addressed with the cross correlation matrix or time-resolved correlation matrix.  See “section 1 Introduction, lines 70-74 and 77-80”

Comment: The EEG pre-processing is qualitatively and not quantitatively described. Since the pre-processed EEG is the input of the authors’ learning models, I am not able to interpret the results. Moreover, it is not possible for the reader to reproduce the authors’ results without having any information about the pre-processing.

Response: A Table(Table 1.) has been added to give an overview of the EEG datasets. Further, the various steps involved in preprocessing and processing of the EEG data has been added under the respective paragraphs which describe each of the experimental paradigms. Please see Section “4.1 Datasets”

Abstract –

Comment: Lines 2-3:  Over Simplistic definition of Functional Connectivity. See, for example: http://www.scholarpedia.org/article/Brain_connectivity

Response: The definition has been rephrased and updated.

Comment: Line 5: Which methods were used to estimate static and dynamic connectivity?

Response: Correlation Matrix (dynamic connectivity) and Time resolved Correlation Matrix (static connectivity). The cross correlation matrix and the time-resolved correlation matrix are obtained from the editor pipeline option under the connectivity tool in Brainstorm toolbox for MATLAB platform.

Comment: Lines 8-9. Please, describe briefly your dataset giving details on acquisition/experimental protocol

Response: A table(“Table 1.”) containing the steps involved in collection/recording of the EEG datasets. Further steps involved in data preprocessing/processing have been added under “section 4.1 Datasets” for the respective EEG datasets.

Original Articles published using the three datasets from Openneuro have been cited and added to the “Reference” section

Comment: Line 9: RNN was not defined before (to define in line 4?)

Response: Line 4: “Recurrent Neural Networks (RNN)” has been added to the text

Comment: Lines 13-14: The conclusion statement is not clear to me.

Response: The Conclusion statement has been rephrased and realigned to explain the concluding statement of this study.

“two different types of correlation matrices: static and dynamic: time-resolved correlation matrix,” 

“Recurrent Neural Networks (RNN) model” provided higher accuracy (74-88%) than the other three models (0-78%).

Introduction –

Comment: Line 19 “:” instead “;”

Response: Has been addressed.

Round 2

Reviewer 1 Report

My comments have been adequately satisfied. However, I further recommend references to the tables to be included as well.

Author Response

We thank the reviewer for his/her valuable inputs in improving our manuscript. As per your recommendation, the tables and figures have been referenced in the main text along with overall improvements to the manuscript. Once again we appreciate the time and energy dedicated by the reviewer in providing his/her valuable inputs. 

With warms regards,

Gautam Kumar B.

Reviewer 2 Report

The authors have partially improved the manuscript, but in my opinion:

  1. The EEG pre-processing and processing is still not described in details to reproduce the results. 
  2. Functional connectivity that is a complex issue and problem, it is still treated as if it were much simpler than it really is.

-----------------------------------

1. Despite Table 1 has been added, there are no details on the kind and order of the filters, if it was avoided phase distortion... Moreover, info in column 4 are not chronologically ordered: I suppose that the authors, first, have checked impedances, second, acquired @1000 Hz (Is the sampling frequency reported or the downsampled frequency after filtering? Not clear), third, filtered their data.

E.g.,

Line 203: Basic filter option --> please, report kind and order of filtering

Lines 203-204 -- 214 --> How many ICs were computed? How many discarded?

Lines 242-243 --> How many epochs (in percentage) were discarded?

Lines 245-247 --> Please, describe briefly the pipeline of “connectivity option”

2. Lines 70-74 and 77-80 are generic and do not inform/alert the reader about all the drawbacks of computing functional connectivity from EEG data (see e.g., https://doi.org/10.3389/fnsys.2015.00175 ). Since the authors have chosen the simplest method to compute a relationship between EEG signals, they should highlight in their discussion that their aim was not to estimate the true causal connections among the brain signals, but to test a classification algorithm.

Minor

Figure 3: please, enhance the resolution of the text

Figure 4: please, increase font size

In figures caption, please, briefly describe the content of the figures.

Author Response

Dear Reviewer 2,

Big Data Cognitive Computing,  

Special Issue- Knowledge Modelling and Learning through Cognitive Networks,  

Thank you for giving us the opportunity to submit the draft after the second round of revision of our manuscript titled “A Novel Approach to Learning Models on EEG Data Using Graph Theory Features-A Comparative Study” to the Special Issue of Big Data and Cognitive Computing titled “Knowledge Modelling and Learning through Cognitive Networks”. We appreciate the time and effort that you have dedicated to providing valuable feedback on our manuscript. We are grateful for the insightful comments on our paper from the first round and for helping us in communicating the work in a coherent form. We have been able to incorporate changes to reflect most of the suggestions provided by the reviewers. We have highlighted(orange) the changes within the manuscript. 

The two major comments raised, which are listed below:

  1. "The EEG pre-processing and processing is still not described in details to reproduce the results.

Response: The preprocessing steps involved in each of the datasets have been mentioned in chronological order in Table 1. Further, under section 4. Materials and Methods subsection 4.1 Data sets the preprocessing and processing steps implemented for each of the datasets is described.

  1. Functional connectivity is a complex issue and problem, it is still treated as if it were much simpler than it really is.

Response: We agree with the reviewer that functional connectivity using EEG signals is a complex issue and has key challenges that need to be addressed. With respect to this study, the objective is to use methods that are simple and have short compute time for calculating the functional connectivity and use the matrices as input for classification algorithms. The key problems have been highlighted and discussed briefly, please see lines 118-138, 185-193

A point-by-point response to the comments and concerns is provided in the attached PDF.  
